# New Perspective on Why Women Live Longer Than Men: An Exploration of Power, Gender, Social Determinants, and Capitals

**DOI:** 10.3390/ijerph18020661

**Published:** 2021-01-14

**Authors:** Fran Baum, Connie Musolino, Hailay Abrha Gesesew, Jennie Popay

**Affiliations:** 1Southgate Institute for Health, Society & Equity, Flinders University, Adelaide 5042, Australia; connie.musolino@flinders.edu.au; 2College of Medicine and Public Health, Flinders University, Adelaide 5048, Australia; hailushepi@gmail.com; 3Epidemiology, School of Health Sciences, Mekelle University, Mekelle 231, Ethiopia; 4Division of Health Research, Faculty of Health & Medicine, Lancaster University, Lancaster LA1 4YW, UK; j.popay@lancaster.ac.uk

**Keywords:** gender inequities, health inequities, life expectancy, social determinants, Ethiopia, Australia, power, Bourdieu, patriarchy

## Abstract

Background: Women live longer than men, even though many of the recognised social determinants of health are worse for women than men. No existing explanations account fully for these differences in life expectancy, although they do highlight the complexity and interaction of biological, social and health service factors. Methods: this paper is an exploratory explanation of gendered life expectancy difference (GLED) using a novel combination of epidemiological and sociological methods. We present the global picture of GLED. We then utilise a secondary data comparative case analysis offering explanations for GLED in Australia and Ethiopia. We combine a social determinant of health lens with Bourdieu’s concepts of capitals (economic, cultural, symbolic and social). Results: we confirmed continuing GLED in all countries ranging from less than a year to over 11 years. The Australian and Ethiopian cases demonstrated the complex factors underpinning this difference, highlighting similarities and differences in socioeconomic and cultural factors and how they are gendered within and between the countries. Bourdieu’s capitals enabled us to partially explain GLED and to develop a conceptual model of causal pathways. Conclusion: we demonstrate the value of combing a SDH and Bourdieu’s capital lens to investigate GLED. We proposed a theoretical framework to guide future research.

## 1. Introduction

Life expectancy at birth is widely recognized as a summary measure of mortality that allows the extent and trend of health inequalities to be compared within and across societies [1,2]. Since 2006, in all countries in the world, women have lived longer than men, but paradoxically report more illness than men [3]. It is also true that women are more likely to live in poverty and have less control in their lives than men and, again, paradoxically, people living in poverty and those with less control are more likely to die earlier than those who don’t [4]. Despite much scholarship, there is still no convincing answer as to why the interactions between gender and access to material resources and power do not produce the expected pattern of life expectancy between men and women [5,6,7,8]. Explanations lie in both biology and social determinants, including dominant gender roles and practices [9]. Research has identified significant gender differences for many common diseases and disorders. However, while biological differences may account for some of the gendered life expectancy differences (GLED), the size of these variations, and the ways in which they change across time and geography in response to socioeconomic changes and public policy contexts, demonstrate that they cannot be accounted for by intrinsic biological differences alone [5,10]. In this paper we therefore adopt an exploratory and comparative approach to analysis of secondary data sources to offer new perspectives on the female advantage in life expectancy. The approach combines a macro social determinants of health lens with a micro focus on how gendered patterns of Bourdieu’s capitals play out in women’s and men’s daily lives. We focus on life expectancy only, given space and data limitation considerations, but we recognise that further insights could be gained by also examining differences in women and men’s experience of illness.

### Background

Differential control over power and resources and unequal daily living conditions were central to the World Health Organisation’s (WHO’s) Commission on the Social Determinants of Health’s [11] explanation for the existence of health inequities. However, the question of why women now live longer than men in all countries in the world despite possessing less power, control and resources was not addressed. Earlier, in the 1970s and 1980s, when gender life expectancy gaps were greatest, especially in the USA, intensive research attention in high-income countries was directed at understanding the drivers of these gaps [12,13,14,15]. However, the explanations offered only accounted for part of the difference. For example Rogers et al. [16] examined U.S. data and found that while co-variates explained some of the difference (especially men’s higher smoking rates) there was still a 62 per cent unexplained gap in life expectancy between women and men. Explanations are also likely to be different between countries, particularly countries with different levels of national income and development.

The Commission on the Social Determinants of Health [11]‘s framework recognises the complexity inherent in accounting for the diverse patterns of population health outcomes that emerge from the dynamic interrelationships between biology, structural determinants and individual and collective agency. However, much research fails to rise to this challenge. For example, while some epidemiological studies have considered reasons for GLED, most control for, rather than seek to explain, the impact of different social variables, despite a call for more sophisticated epidemiological studies [17]. More than three decades ago, Macintyre [18] noted that British health research generally treated social variables, such as occupational class, gender, marital status, age, ethnicity and area of residence, as separate strands rather than examining the interactions between them. This continues to be a problem.

To address this, we build on the work of scholars who have been challenging reductive and positivist approaches to gender and health [19,20,21], but until recently have remained somewhat on the margins of global health investigations into gendered differences in health outcomes [10,22,23,24]. This body of work adopts a sociological and gendered lens locating the conditions “of daily life in the context of patriarchal structures and ideologies” [25] (p. 86). Patriarchy at its core comprises a system of power and a gender order [26]. Based on people’s conformity to dominant masculinities and femininities, patriarchy upholds a gender binary which is hierarchical and has many implications for women and men’s health. For example, Hart et al. [27] observe that historically notions of the gender binary and beliefs about the gendered nature of health have produced stereotypes that align femininity with vulnerability, weakness and hysteria, and masculinity with control, stoicism and strength. This positions masculinity as favourable for health yet, as demonstrated by many studies, gender-stereotypical behaviours for women are often protective of health, whereas stereotypically masculine behaviours are often more detrimental (risk taking, delaying help seeking, violence) [27]. Furthermore, gender scholars challenge dichotomous and static understandings of gender and sex [20,28,29]. Connell and Messerschmidt [30] conceptualised the term “hegemonic masculinity” to highlight that there exist different types of masculinity, but that a heterosexual masculinity is currently the most dominant and socially acceptable. Those who conform to this form of masculinity are rewarded by society through access to power and capital, and those who do not are disadvantaged, including men from marginalized groups, by class or race or sexuality. Gender and patriarchal structures are therefore implicated in patterns of disease and injury within and between countries and also how these patterns change over time (e.g., women’s increasing smoking rates in some countries are partly attributed to tobacco industries gendered marketing [31]). As another example, Bates et al. argue “normative masculinity can, for example, function in the interests of the state and elites by fostering tolerance among men (predominantly of lower SES) for harmful exposures in the service of war or dangerous work” (p. 1003). Therefore understanding how “gender is embedded in unequal power relationships” [24] (p. 238) and how gendered norms, practices and structures operate to protect or damage health [28] is central to understanding differences in unequal health outcomes.

We aim to combine theoretical insights from gender theory with insights from Bourdieu’s [32] theory of the relationship between economic, cultural, social and symbolic capitals in order to explore how gendered power dynamics relate to social position [28]. Over the last few decades, researchers have begun to apply Bourdieu’s capitals to health and gender experiences and inequities as the theory accommodates both individual agency and social structures [33,34,35,36,37,38]. Turner [39] suggests that Bourdieu’s conception of social capital explains how social structures can influence quantity and quality of a person’s social relationships, which play an important part in the maintenance of health, and, at the same time, provide resources for recovery from illness. Others have stressed the importance of cultural capital in explaining health inequities [37,38]. For example, Oncini and Guetta [34] look at the interplay between women’s increasing cultural capital and increasing smoking rates in Italy. They found women adopted unhealthy smoking behaviours as their cultural resources (education level, books read, participation in cultural activities) increase. This meant understanding the gendered processes of cultural capital in Italy was a better predictor of smoking behaviour than socioeconomic status. Bourdieu’s framework of capitals promises to be highly relevant to understanding differences in GLED for several reasons. Firstly, it allows examination of differences in LE to consider a wide range of causal factors beyond the economic. Secondly, it provides a structural framework to understand how individuals’ practices affect health. Thirdly it does this in a way that avoids victim blaming because it emphases the resources people have rather than those they lack. Because of this, Pinxten and Lievens [40] note that it enables a theory of privilege rather than one of inadequacy.

In this article, we present an exploratory analysis using a novel combination of existing social science theory drawing on a social determinants of health lens informed by gender theory and Bourdieu’s capitals, to examine the ways in which power and structural factors influence everyday life and individual agency and whether this perspective is helpful in understanding GLED.

## 2. Methods

Our research is based on analyses of secondary data sources and comprises two main dimensions: analysis of gendered life expectancy differences around the globe; and a comparative case analysis of these differences in two countries, Australia and Ethiopia.

### 2.1. Global Analysis of Gendered Differences in Life Expectancy

We examined gendered differences in life expectancy (GLED) for all countries in the world based on data from the World Bank. We categorized the GLED as very low (<3 years), low (3–<5 years), moderate (5–<7 years) and high (>7 years) in order to look at the distribution of differing GLEDs according to levels of development.

### 2.2. Comparative Case Analysis

As Bartley [41] (p. 147) notes, “the problems involved in understanding gender differences in health… seem to be most helpfully analysed by international and comparative studies”. We therefore undertook a comparative analysis of two countries well-known to one or other of the authors: Australia and Ethiopia.

Given the novelty of the examination of GLED and the anticipated challenges of our reliance on diverse secondary data sources, we decided we would only be able to manage a robust albeit preliminary analysis of two cases. Our selection criteria for the two were a combination of theoretical and methodological. Theoretically we decided we would gain greater analytical advantaged by comparing two countries with: (i) diverging national income levels: one high and one low-income; and (ii) higher than expected overall LE given level of Gross Domestic Product (GDP). Methodologically, we selected two countries for which relevant secondary data sources (e.g., World Bank database) and published literature and/or data on policies, and political and historical context was readily available to us and members of the author team had knowledge of these sources. We note that sex disaggregated data are more readily available in Australia than in Ethiopia, where the data comes primarily from population health surveys rather than routine national data collections.

#### 2.2.1. Theoretical Framework for the Comparative Case Analysis

Our comparative case analysis is framed by a novel combination of existing social science theories. The social determinants of health perspective offered by the Commission on the Social Determinants of Health [11] considers both the distribution of power and resources and conditions of everyday life, and we examine these through a gender lens, to determine the ways in which patriarchy operates. To assist our analysis of power and resources we draw on Bourdieu’s theories of the interplay between different forms of capital:Economic capital, which refers to material and financial resources.Cultural capital, including education, literacy skills and possession of items of distinction.Social capital, which is gained from relationships and networks and can be tight knit bonding capital or bridging capital, extended across class, groups and “agencies”.Symbolic is the power that comes from the deployment of any form of capital and is associated with honour, prestige, or recognition. It symbolises the value that a person holds within a social field [32].

We examine how possession of these four capitals differentiate the experiences of men and women in the two countries and ways that either expand or contract individuals’ life chances. We then consider how possession of differential levels and combinations of these capitals amongst women and men may affect health outcomes and so translate into gendered life expectancy differences.

#### 2.2.2. Ethics

This study does not require ethical clearance as data were obtained from published sources.

## 3. Findings

### 3.1. Global Picture of Gendered Life Expectancy Difference (GLED)

Our analysis of World Bank data indicates that in all countries, women have longer life expectancy than men. The range of difference was from only 0.7 years in Bhutan to 11.2 years in Syria. When considered by income level, there is a gradient of within country GLEDs among low-, middle- and high-income countries (Table 1). The difference between men and women’s life expectancy is least in low-income countries (3.8 years) and highest in high-income (5.2 years). The within country GLEDs are less than those across countries. Thus, the most “advantaged” groups—women in high-income countries—gave a life expectancy that is 21.76 years more than the group with the lowest life expectancy—men in low-income countries.

Out of 194 countries and territories, 30 countries have very low, 66 countries have low, 41 countries have moderate, and 25 countries have a high GLED. While members of the Arab League and other Muslim countries dominated the very low GLED group, none of the high-income countries in North America, Europe or Australasia are in this category. Bhutan and Guinea have the lowest GLED with 0.7 and 1.2 years, respectively. Comparatively, East Europe and Latin American countries have high GLED but only two African country have high GLED: Seychelles and Eswatini (Swaziland). Syria, Belarus, Russia and Ukraine have the highest GLED with 11.2, 10.2, 10.1 and 10 years, respectively.

The trend over the twentieth century in high-income countries has been for the gender difference in life expectancy to increase up until the 1970s and thereafter to decline [42]. However, the patterns were significantly different in high-incomes countries compared to low- and middle-income countries. In a study of 17 high-income countries Thorslund et al. [42] identified three phrases the GLED had gone through, including: a period with a slight advantage for women (Phase A); a period with a rapid increase in women’s advantage (Phase B); followed by a period when the GLED decreased (Phase C). All countries seem to pass through all three phases, with the exceptions of Japan and Spain that did not show a decrease (Phase C). They attribute the widening gap during most of the 1900s to a slower mortality decline for men than women, which previous studies attributed to behavioural risk factors (e.g., smoking). Cigarette smoking among men had the greatest contribution to the widening of GLED, which led to higher mortality from lung cancer and heart disease. Gender gaps in cardiovascular disease mortality had also emerged as the main component GLED from all causes in industrialized countries. In low- and middle-income countries, in contrast, for much of the 20th century, men had an advantage over women, but that pattern reversed in the last decades of the century and into the first decade of the twenty-first century [6].

### 3.2. Comparative Case Analyses: Australia and Ethiopia

In this section we compare Australia and Ethiopia by firstly looking at the different epidemiological profiles. We then examine legal and political differences in the position of women and men in the two countries and access to health care and consider how these determinants could be linked to GLEDs in these two countries. Finally, we use the lens of economic, cultural, symbolic and social capitals to add to our explanation for GLED within and between Australia and Ethiopia.

#### 3.2.1. Epidemiological Perspective

Australia and Ethiopia provide contrasting cases in terms of overall life expectancy (Figure 1) and in terms of the GLED (Figure 2). Australia is a high-income country and this is reflected in its total life expectancy at birth, which in 2017–2019 was 82.9 years: 85.0 for women and 80.9 for men [43]. Ethiopia is a low-income country, which has made significance gains in life expectancy over the past three decades (Figure 1). In 2018, Ethiopia’s total life expectancy was 66 years: 68.2 and 64.4, respectively, for women and men [44]. From around 1980, there has been a marked change in the GLED trend in Australia as it decreased from an approximate 7-year advantage for women in the 1980s to just 4.1 years in 2017–2019. In the last ten years, the increase in life expectancy has been greater for men (1.6 years) than for women (1.1 years) [43]. In Australia, men’s mortality began reducing at a faster rate than women’s from the 1980s, for most causes. Declines in mortality from specific causes, especially among men, have been attributed by Adair and Lopez [45] to public health interventions, including those for reducing tobacco smoking and improving road safety. In Ethiopia, in contrast, the GLED remained very stable from the 1960s with women living an average of 3 (2.7–3.8) years longer than men until, around 2005, since when the gap has slowly widened [44]. These contrasting trends have seen a convergence of the GLED in these two very different countries.

Aside from gender there are other significant differences in each country [45]. Most starkly in Australia in 2015–2017, life expectancy at birth for Indigenous Australians was estimated to be 71.6 years for males and 75.6 years for females: though the GLED, at 4 years, was very similar as the population as a whole. For Indigenous peoples living in remote areas LE was even lower at 69.6 for women and for 65.9 men [46], as was the GLED (3.7 years). Comparing the area with the longest life expectancy (North Sydney and Hornsby) (also one of the most affluent areas in Australia) with the lowest area (Northern Territory (NT) Outback) indicates a 12.3-year gap. The GLED for North Sydney and Hornsby is 2.4 years compared to 4.4 years for the NT outback. A study of life expectancy, gender and socioeconomic status in Australia between 2001 and 2012 found that gender gaps increased with increasing socioeconomic disadvantage [47].

Though there are fewer sources in Ethiopia, the data that are available point to geographical and socioeconomic differences in life expectancy overall. Rural residents were calculated to have a mortality rate of twice that for their urban counterparts [48,49]. There is also a clear socioeconomic gradient in Ethiopia, with life expectancy ranging from 53.4 years in the lowest wealth quintile to 62.5 years in the highest quintile—an absolute difference of 9 years [48]. Unfortunately, these data are not disaggregated by sex.

Table 2 provides data on the ten top causes of death for women and men in the two countries. Australian figures are based on national statistics. National statistics in Ethiopia do not provide a gender breakdown of cause of death, so these data are therefore drawn from a verbal autopsy study in North Ethiopia conducted by Melaku et al. [50]. In Ethiopia, infectious disease is the main cause of death for men and women, and external causes of death (which include accidents, self-harm, assault and lack of food) are also high for both men and women. In contrast, in Australia, whilst the top four causes of death are the same for men and women the order is different. For men, cardiovascular and lung cancer are the most common, and suicide is also significant. Dementia (a disease of old age) is the most common cause of death in women followed by coronary heart disease. Though work injuries are not a major cause of death they are strongly gendered with 90 per cent of deaths due to injuries at work were men and in 2011/2012, 176 of 190 people killed at work were men.

##### Measures of Gender Equity

Table 3 provides the United Nations (UN) Gender Inequality Index (GII) and the UN Gender Development Index (GDI) for Australia and Ethiopia from 2000. The higher scores on the GII for Ethiopia indicate higher inequality between men and women and the lower scores on the GDI are consistent in that they record less gender parity. The trend for both countries is towards greater gender equity regarding empowerment and human development though improvements in the GDI appear to have stalled since 2005 in Australia and 2015 in Ethiopia. In Ethiopia, the trend to greater gender equity has been associated with a small increase in the GLED, whereas in Australia, the small rise in gender equity was correlated with a reduction in the GLED. It should also be noted that in 2018 Ethiopia’s gender-related development index (GDI) was 173rd out of 181 countries [52], suggesting Ethiopian women continue to be severely disadvantaged [52]. We note the GII and GDI contain mortality data, but they are not solely based on these measures.

#### 3.2.2. Sociological Perspective Extending Legal and Political Rights: for Women

Women (except for Indigenous women) have had full suffrage in Australia since 1902. For women in many high-income countries, the Second World War was a turning point for women’s equality in Australia as women increasingly became wage earners [53]. Today, Australia has sexual discrimination legislation that is designed to prevent workplace discrimination on the basis of gender. Women can own property on the same terms as men. The marriage bar that legally prevented women from working in education after marriage was lifted in 1956, and access to the contraception pill was allowed in 1961 (initially for married women only). Other wins for women’s rights included the right to equal pay in 1969, no-fault divorce in 1975, the Sex Discrimination Act 1984, signing to the UN Convention on the Elimination of Discrimination Against Women (CEDAW) in 1980, and access to safe abortions is legal, although there are still several restrictions. In terms of political representation, state and federal governments have appointed women’s advisors who have worked to increase gender equity since the 1970s. More women now sit in federal parliament, but they remain underrepresented in leadership roles. Only 26.1% of federal government cabinet ministers were women in 2019 [54], and in 2018, there were only 57 female commonwealth justices and judges compared to 100 men [54].

Ethiopia’s recent political history has seen the replacement of a feudal (before 1970s) and autocratic government (before 1990s) by a more democratic one (1991–2018). This change has seen the country ratified the Convention on the Elimination of All Forms of Discrimination against Women (CEDAW) in 1981 [55]. Additionally, it signed the 2003 Maputo Protocol, which guarantees women’s right to participate in political processes equally with men [55]. Ethiopia has also introduced several national policy and legal reforms to close the gender equity gap. For example, a national policy for women and a Ministry of Women’s Affairs were established in 1993 and 1995, respectively [55]. Political representation of women in Ethiopia’s house of federation was thirty-nine per cent in 2019, compared to a Sub-Saharan African average of twenty-four per cent [56]. Similarly, in 2013 the representation of women was more than thirty per cent in several regional states councils and forty-eight per cent in the region Tigrai [57]. However, the national policy for women and the Ministry of Women’s Affairs are under-resourced, which substantially limits their abilities to implement gender equity measures and the key political, leadership positions and decision making processes remain dominated by men [55].

Ethiopian women have equal land rights with men. Since 2000, a photo and signature from both spouses have been required to obtain a property ownership certificate [55]. Additional women’s legal rights were introduced in the mid-1990s, including paid maternity leave, and other employment rights [55,58]. Another sign of the empowerment of women is that, in 2016, eight-one per cent of women reported that they made decisions about their own health and seventy-eight per cent reported making large household purchases. However only forty-five per cent reported having control over their sexual relations [59].

In summary, in Australia the extension of women’s legal right and access to political power has occurred at the same time as the GLED has declined primarily because of a sharper decline in male mortality. In Ethiopia the gradual improvement in women’s rights in the past two decades has seen a rise in the GLED, albeit very small.

##### The Role of Health Systems

Australia has a universal publicly financed health care system. A women’s health movement has been successfully advocating for better access to services for women for thirty years [60]. They have access to abortion, and (albeit with some restrictions) and high quality maternal and postnatal care has been available for a long time. The health and education systems have universal access and in general access has been equitable for women for many decades. However, significant gaps in services exist for people in rural areas, people from a non-English speaking background and for Aboriginal and Torres Strait Islander populations [61]. There are also service access gaps for people in lower socioeconomic groups for whom cost is a barrier and these are greater for women than men. For example, women are more likely to report (1 in 25) than men (1 in 40) not seeing a GP when needed because of cost reasons at least once in the previous 12 months [62]. Thus, gender and material disadvantage interact to limit access to health services in Australia.

Ethiopia adopted a health care financing strategy in 1998 [63,64] and a community based health insurance scheme (CBHI) [65,66] in 2011 to improve the quality, equity and accessibility of health services [63,66]. Despite some improvement, however, inequalities persist. For example, while over eighty per cent of the population in Ethiopia live in rural areas, there are better ratios of doctors, nurses, midwives, and health officers [67], and approximately ninety per cent of hospitals are in urban areas [68]. There are also low rates of access to reproductive health services and some differences, according to household structure. For example, access to contraceptive services was sixteen per cent for female headed compared to twenty-three per cent for male headed household and the coverage of skilled attendance at birth for female headed and male headed household was thirteen per cent and twenty-one per cent, respectively [69]. Female headed households are also about three times more likely to take up community based health insurance scheme (which improves access to services) than their male counterparts [70].

##### Exploring Gendered Everyday Practices through the Lens of Bourdieu’s Capitals

Our cross-country analysis has so far revealed significant improvements in women’s rights and their access to resources and services over time in both countries with improvements occurring more recently in Ethiopia. However, we have also shown that these improvements have emerged alongside continued gendered inequalities in the power and resources that structure participation and influence in different aspects of society. As Popay and Groves [25] (p. 85) argued it is vital to consider how “the gendering of the social, cultural, material and ontological dimensions of daily life shape women’s and men’s health experiences”. In this section we therefore examine the ways in which Bourdieu’s different forms of capital shape the everyday practices of men and women in Australia and Ethiopia and consider how these differences may translate into GLEDs.

##### Economic Capital

In both countries there are gender gaps in the distribution of employment, income and wealth. In Australia in 2018–19, two-thirds of women (67.4%) and more than three quarters of men (78.5%) aged 20–74 years old participated in the labour force [71]. In comparison, amongst parents with a child under six, these gender differences widened considerably to 64.2% of women and 94.6% of men. Women are also more likely than men to be working part-time and in casual jobs, and to be underemployed. Over the same period, 94.9% of primary parental leave (paid or unpaid) was taken by women. Australian women also spend 80.8% more time on unpaid household work each day than men [72]. Costs of early childhood education in Australia, among the highest in the Organisation for Economic Co-operation and Development (OECD) at around twenty-seven per cent of families’ incomes [73], is seen as a barrier to paid employment for low and middle-income mothers.

Gendered income and wealth gaps also persist in Australia. The full-time total remuneration gender pay gap in 2017–2018 was 21.3% [74]. Over the decade up to 2018 the average adult weekly earnings for women has remained around 69% of that for men. Additionally, the majority of sole parents are women (83%) [75] and are more likely to be living in poverty [76]. Reflecting these inequalities in life time earning, thirty-four per cent of single women over 60 in Australia are living in permanent poverty compared to twenty-seven per cent of single older men and twenty-four per cent of couples in these age groups [77]. Superannuation balances are an indicator of inequalities in wealth accumulation and the median balance at official retirement age was approximately a third less for women than men in 2017–2018 [74].

The picture for gender inequalities in economic capital is similar in Ethiopia. The 2016 Ethiopian Demographic and Health Survey indicated that only twenty per cent of women in rural areas compared to 52 per cent in urban areas were employed in the preceding 12 months, while the figures for men were ninety per cent in rural areas compared with 80.5% in urban areas. Women in higher socioeconomic groups are more likely to work than those in the lower quintile (48.7% compared with 24.4%), whilst there is very little difference for men (84.1% compared to 86%) [78]. A study [79] comparing the situation of women and men between 2005 and 2011 found that despite improvements men were more advantaged than women on a number of socioeconomic indicators. For example, while the unemployment rate was steady at 64% for women over the period for men it had increased but only from 12.4% to 22%. This study also found significant differences between women. For instance, in 2011 women in Addis Ababa were 6.4 times more likely to be employed than women in rural Tigray up from only a 1.5 difference in 2005. As in Australia, women in Ethiopia also carry a higher burden of, unpaid, household role responsibilities than men, including family care, food production, and preparation. For example, thirty per cent of women, compared to eighteen per cent of men, spend time collecting firewood or water [80].

In both countries, women are under-represented in senior positions in the private sector. For example, the Ministry of Trade in Ethiopia reported that only seventy per cent of all businesses were owned by women [80]. In Australia, 82.9% of senior leadership positions in the non-public sector are taken by men. Available data on the distribution of economic capital indicate that daily life in both countries is characterised by gendered structures which give most economic power and resources to men.

Australian women have gained more economic power over the last 50 years, but they are still more likely to live in poverty, to earn less, have less wealth and are much less likely to hold senior positions in the private sector than men. Over the period in which Australian women have made economic gains GLED has diminished, as men’s life expectancy has increased at a faster rate than women. Some commentators have suggested that women have experienced greater stress as a result of juggling work and family life that is likely to have adversely affected their health especially their mental health [81]. Given that men continue to hold more powerful positions in workplaces, as women have entered the labour market their status is more likely to be lower than men’s and lower social status has been recognised as detrimental to health [82].

In Ethiopia, the last thirty years have been a period in which women have gained more economic power, yet this does not appear to have made much difference to the life expectancy gap. The relatively rapid rise in life expectancy for both genders likely reflects the increasing social and economic development experienced by the population as a whole and greater political stability in the country.

Other studies have found that the gender division of labour has typically placed men in more physical and dangerous work environmental conditions and higher risks of accidents [83,84]. In both Australia and Ethiopia, men are more likely to die because of external causes, but in Ethiopia, this cause is also significant for women, suggesting different gender patterns for these risks between the two countries. In Ethiopia, more women work in rural areas than urban areas and are therefore more likely working in physically risky jobs [78]. Additionally, as we reported earlier, infectious diseases, typically associated with poorer living conditions, continue to be the most significant cause of death for both men and women in Ethiopia. Overall, however, our analysis of gendered inequalities in access to economic capital would more easily support a situation where men live longer than women (given the extensive empirical literature on the link between income and wealth and health [85,86,87,88])—which is not the case in either country. Thus, other factors must be at play.

##### Cultural Capital

Shim [38] notes that Bourdieu conceptualizes cultural practices and products of all kinds—Ranging from styles of dress, eating habits, verbal skills, scientific knowledge, educational credentials, and so on—as forms of cultural capital. Cultural capital is strongly linked to economic capital as literacy, education and qualifications provide access to employment. In Australia, the education system does not discriminate between men and women. Differences in education access, literacy rates and attainment between men and women today are minimal, although women are more likely than men to have attained a bachelor’s degree or higher qualification. The increase in women’s education qualifications is likely to have been beneficial for both their male and female children and will have contributed to the overall gains in life expectancy. Shim [38] proposes the concept of cultural health capital, which she suggests influences healthcare encounters in the U.S. Many women in Australia may also possess more of this capital and so benefit more from health care than men.

In Ethiopia, women’s education has improved [89] with females’ national gross enrolment increasing from 26 to 97% between 1997 and 2015 [90,91]. As a result, women’s literacy has increased from 27 to 49% over the same period [65,66,92,93], increasing opportunities for employment. Men’s literacy rates have also improved over the same period from 36 to 59% [94]. However, only twenty-seven per cent of the university population are female and twenty-five per cent of women drop out before graduation due to economic factors, misconceptions about girls’ academic ability, and gender bias of institutions [95].

Female gains in aspects of cultural capital have been significant in both countries but most of these gains reflect what has been referred to in a recent United Nations Development Programme report on gender inequality as basic capabilities (education, right to vote) and less in enhanced capabilities, for example, through access to higher status, better paid employment and more powerful positions in the business and corporate world [96]. Arguably, this underpins their relatively disadvantaged position in access to economic capital compared to men.

##### Social Capital

Turner [39] maintains that Bourdieu’s conception of social capital is an important concept to understand health in contemporary society. He argues (p. 4) that “the quantity and quality of a person’s social relationships and social networks play an important part in the maintenance of their health, and at the same time provide resources for their recovery from illness”. Bourdieu’s conceptualisation of social capital was concerned with the ways in which the density of social connections can increase an individual’s power, status and resources it gives rise to. Social capital appears most likely to work in favour of women’s health. Australian women are more socially connected than men and have more people they can call on for help. However, the relationship between social capital and health outcomes are complex. For example, Denton and Walters [97] found that social support was more strongly associated with better health for women than for men, but in their study of social capital Berry et al. [98] found women reported greater community participation and social cohesion than men, yet worse mental health. Another study [99] found that among Australian adults aged 25–44 living alone more men experienced loneliness (39%) than women (12%) and in a meta-analysis of research Holt-Lunstad et al. [100] showed that the risk of premature death associated with social isolation was twenty-nine per cent and for loneliness twenty-six per cent.

Australian women are also more likely to participate in social and community groups than men but research [101] suggests such participation often depends on access to sufficient economic capital. Additionally, the effect of more women entering the workforce may be that their opportunities to build social connections in their local communities have declined and not been entirely replaced by work social contacts. Trade union membership in Australia has declined from 1992 to August 2016, from 40 to 15% but slightly more women are now in trade unions than men (16% compared to 13%), meaning women are slightly more likely to benefit from the social capital stemming from union membership [102]. From the 1960s, there has also been a strong women’s movement in Australia, which has built social capital among women and advocated for women’s rights [60,103].

Data on social capital are much more limited in Ethiopia, but a study [104] of the link between subjective wellbeing and social capital in rural Ethiopia found that gender did not predict this link. Civil society organizations in Ethiopia, which were oppressed under the feudal monarchy and military junta, increased substantially following the transition to more democratic government in 1991 [105]. This is likely to have benefited women and men but many of these organizations contributed specifically to gender equality and women’s empowerment. An example is the Women’s Association of Tigray (WAT), which advocates to prevent violence against women and girls [106]. Women’s participation in trade unions also increased from two per cent in 2005 to six per cent in 2016 with most members living in urban areas [59].

The CSDH Gender Knowledge Network noted that women function as the “shock absorbers” for families, economies and societies through their caring responsibilities and they represent an important resource for families and communities in both Ethiopia and Australia. It is not easy to determine, however, in either society, whether the benefits of the greater social capital women accrue compared to men for health and wellbeing outweigh the pressures of providing informal care. There are well-recognised mental and physical health hazards associated with informal care and women are more exposed to these [107,108].

##### Symbolic Capital

Symbolic capital arises from the prestige a person holds in their society or community. Individuals accumulate symbolic capital through forms of recognition and status. It is more difficult to measure empirically than other forms of capital, but it may hold clues for explaining the GLED. Holding more economic resources confers more symbolic capital and therefore, in both countries, builds the symbolic capital of men more than women. However, the pattern differs greatly across groups with more or less economic resources so that social class may be a greater predictor of levels of symbolic capital than gender. The distribution of symbolic capital may also reflect traditional practices and values in a society. In Ethiopia, for example, married women with children are accorded considerable respect, recognition and status in their communities [109,110]. This is likely to be beneficial for health, and while Ethiopia has become more urbanised, a process that often undermines traditional cultural practices, this has not yet happened to any significant degree in the country. In Australian women have gained more independence and divorce rates have increased. The introduction of a single parent pension in the 1970s accorded these women more economic freedom. Although, initially stigma was attached to both divorce and single parenthood [111] and may have resulted in women in these positions holding less symbolic capital especially in an era when politicians disparage welfare recipients including single parents [112]. Furthermore, as the final report of the CSDH Gender Knowledge Network [113] noted women may support oppressive norms because this gives them status and thus symbolic capital, despite being painful (female genital mutilation, cosmetic surgery) or dangerous to their health.

Social norms are one way of assessing symbolic capital and the UN gender social norms index, which measures the extent of patriarchal attitudes indicates significant differences between Australia (2010–2014 data) and Ethiopia (2005–2009 data) [96]. In Australia, forty-six per cent have one gender bias compared to eighty-five per cent in Ethiopia, while fifty-four per cent have no bias in Australia and fifteen per cent have none in Ethiopia. This difference like those in the GDI and GII (Table 2) suggest that Ethiopia is a more patriarchal society than Australia despite some economic and cultural gains for women. Power in a patriarchal society is strongly gendered and results in men holding more political and economic power.

### 3.3. Gendered Social Health Practices

The four types of capitals, singly and in combination strongly shape the social health practices that have a direct impact on mortality. For example, economic capital dictates the extent to which healthy practices can be afforded, cultural capital is associated with knowledge of the factors that influence health and ability to maintain effective communication with health care providers, social, and symbolic capital influence the acceptability or otherwise of healthy practices. The ways in which Bourdieu’s capitals overlap and affect health has led Schneider-Kamp to propose the notion of “health capital” as a concept able to aid understanding of contemporary trends in health practices and health discourse. The application of this concept provides important new insights into the extent to which people can adopt healthy social practices to influence their health and help to identify the structural factors that impede adopting them. A stark example in our two countries is that in Australia nearly all communities can take the provision of safe drinking water for granted. In Ethiopia, in contrast, its provision for many still depends on individual agency to seek and collect the water—a task that often falls to women.

Australia has routine national data sets on health-related social practices (Box 1), whereas in Ethiopia these data come from the Ethiopian Development and Health Survey (Box 2).

Box 1Australian key social practice risks by gender [114].
Physical activity: 5 in 10 men and 4 in 10 women were sufficiently physically active. (2017–2018).Diet: fewer than 1 in 30 men and 1 in 15 women ate enough fruit and vegetables; 12% of males and 6.4% of females were daily consumers of sweetened drinks (2017–2018).Smoking: men were 1.5 times as likely to smoke daily as women, but smoking rates have declined overall.Weight: 7 in 10 men and 6 in 10 women were overweight or obese (2017–2018).Alcohol use: 1 in 4 men and 1 in 11 women were consuming alcohol at levels placing them at lifetime risk of an alcohol-related disease or injury (decreasing over time) (2017–2018).Violence: 4 in 10 men and 3 in 10 women had experienced physical violence since the age of 15. 4.7% of men and 18% of women had experienced sexual violence since the age of 15 (2016).Self-rated health: males and females were equally likely to rate their health as excellent or very good (2017–2018).


Box 2Ethiopian key social practice risks by gender.
Smoking: 4% of men and 1% of women reported smoking [78].Weight: 21%of women and 12% of men in urban Ethiopia are overweight or obese [78].Sexual violence: the prevalence rate of rape among boys was 4.3% [115], the prevalence rates of life time (age 15+ years) and 12 months physical violence among women in Ethiopia in 2016 were 20% and 15%, respectively; and the life time and 12 months sexual violence rates were 10% and 7% respectively [78].Alcohol use: a systematic review and meta-analysis conducted in 2019 found that the prevalence of hazardous alcohol consumption in men and women was 11.6% and 1.2%, respectively [116].Self-rated health: 10% of participants rated their health as excellent, but women reported significantly lower physical, psychological, and social domains of quality of life [117,118].


The data in Box 1 and Box 2 are not directly comparable and the factors need to be interpreted within the country context. Thus, being overweight is a risk factor for non-communicable disease and the high rates are very important in the Australia context. However, in Ethiopia, overweight is only becoming an issue for urban areas, with food insecurity being more significant in rural areas. Smoking has been a significant issue in Australia and very likely accounts for some of the GLED. At all times, men have been more likely to smoke than women, although smoking among women increased from the 1970s until the rates declined for both groups from the late 1980s [119]. Smoking rates increased amongst women at the same time as they were more likely to enter the paid workforce and to some extent behaviours converged [5]. In Australia, young men are more likely to engage in greater risk-taking and violent practices than women, including smoking, drinking, traffic accidents and homicides [5,120]. Overall in Australia, on most measures, women are less likely to engage in social practices that damage health than men [111,121]. In Ethiopia, some social practices that damage health are less common, for example, smoking prevalence rates are very low for both genders. However, risky alcohol intake is relatively high and more common in men. The data on exposure to violence in Ethiopia are scant but the relatively high deaths from external causes (Table 1) indicates men are more exposed (16%versus 8% for women). The existence of a dominant heterosexual masculinity in both Ethiopia and Australia can mean that men are more reluctant to seek medical or other health care [122] because the status of conforming to that norm does not include the admission of weakness that seeking or accepting health care may involve.

These differences in social practices and exposures that may damage health provide some clues to the gender life expectancy gap. They are shaped by people’s access to social, symbolic, cultural, and economic capitals, and as we have shown, this is strongly gendered. For example, Flood [123] (p. 2386) suggests men who confirm to the belief that men “should be tough, stoic, dominant, daring and in control” are likely to make more adverse lifestyle choices than women. Our data on social practices lend support to this view and the ways in which these are structured are part of the explanation for the longevity of women in both countries. Yet consideration of these practices also points to the intersectionality influences as both men and women’s risk factors differ according to social class and ethnicity. For example, Australian women living in the lowest socioeconomic areas were almost four times as likely to smoke daily as women in the highest socioeconomic areas (19.3 and 5.2% respectively) [114]. After adjusting for differences in age structure, Aboriginal and Torres Strait Islander peoples aged 15 years and over were almost three times as likely as non-Indigenous people to be daily smokers (38 compared to 13%) [124]. There were no significant gender differences.

## 4. Discussion

Our paper has offered a new perspective on framing research seeking to understand the different patterns in male and female life expectancy. These patterns have differed over time and between countries. We noted that women now live longer than men in all countries of the world and that the difference is greatest in high-income countries, even though the gap is not as great in the affluent countries, as it was in the past. The patriarchal countries of the Middle East have very low life expectancy differences and high-income western countries have narrowed their GLED as women have achieved more gender equity. These contrasting cases show the variation in patterns of life expectancy difference and how explaining the patterns needs to be rooted in an understanding of context. The relationship with degree of gender equity is not straightforward. We also pointed to the complex relationship between gender equity, social class and ethnicity. We found that GLED in lower socioeconomic groups is greater than that in higher socioeconomic groups. Using only national average data on gender masks may patterns of inequities that need to be unpacked for full understanding of reasons for GLED.

Consequently, to unpick the reasons for GLED within country context we presented two cases of GLED in Australia and Ethiopia. Both our case countries have had converging GLED difference, with a stronger trend in this direction in Australia. However, the overall trajectory of life expectancy was quite different between the countries in that there was rapid improvement for both men and women in Ethiopia and a much slower increase in Australia, given its already high life expectancy. The comparison shows that there are very different drivers of life expectancy in these countries and that epidemiological and sociological knowledge both offer insights. Our cases also demonstrated that health inequities are multi-facetted and there are intersections based on class, country of birth, race, gender and socioeconomic position, and these relations shift over time. We noted that there are significant LED between people based on rurality and socioeconomic status within countries. We used the social determinants of health lens to shape our analysis and added the forms of capital Bourdieu [32] has proposed as important in the reproduction of inequities, which also shape life expectancy. The consideration of economic, cultural, social and symbolic capitals has proved useful in allowing us to highlight a range of factors that might influence life expectancy and also to consider the interactions between the different forms of capital. The capitals, which accumulate to individuals, are directly influenced by the range of structures that shape their lives. These structures include legal, political, economic and cultural sources of power and resources, which are distributed unevenly in societies. The ways in which people’s lives are structured and the capital they have access to, in turn, affect their social practices and exposure to health risks. Collectively these capitals can be conceptualised as “health capital” [122] and our analysis has shown that health capital is both highly gendered and social position related. Figure 3 provides a representation of how these influences work to affect life expectancies. This framework will be a useful tool for others wanting to examine GLED. It shows that gendered differences in socioeconomic environments shape the ways in which men and women are able to develop capabilities and accumulate the different capitals. This results in strong gender effects on the amounts of the different capitals individuals accumulate and these directly affect GLED. The extent of capitals accumulated also affects the social practices individuals adopt. These practices are strongly shaped by gender norms and also contribute to GLED.

In Ethiopia’s case, while women have gained considerably in terms of economic and cultural capital, their life expectancy advantage has held steady with just a small increase. In Australia, women’s significant increase in both capitals has coincided with a decline in the life expectancy difference. This may be partly because women have adopted lifestyle habits including smoking and drinking as they have left the domestic sphere and been subjected to more work-related stresses. Some clues to explaining these differences may come from the insights from symbolic and social capital. Although women are more likely to work, they do so in less economically valued industries, such as health and social service roles, education and caring, due to them being categorized as gendered roles [125]. Women in Australia suffer from the double burden of paid and unpaid work and so have less time for building and maintaining social capital. In Ethiopia, both men and women have seen the freeing up of civil society since the democratic regime was established, which may have meant increasing social capital for both sexes. At the same time, symbolic capital for women in traditional societies may have declined with the adoption of symbolic capital items more typically associated with western lifestyles. At the very least, the use of the capital framework demonstrates the many factors that need to be accounted for in determining the impacts on lifestyles. Our cases also suggest that different factors may have a different impact on men and women. This supports the work of Denton and Walters [97] whose Canadian multi-level analysis found that being in the highest income category, working full-time and caring for a family and having social support are more important predictors of good health for women than men. Smoking and alcohol consumption were found to be more important determinants of health status for men than women.

Gender equity is a crucial variable in unpacking GLED. The GII and the GDI show that both countries trended towards more gender equity, but this had different correlations with the direction of the life expectancy gap suggesting as other researchers have found that the relationship between life expectancy and gender will differ depending on stage of development of a country [5]. An Australian study found that gender equity was associated with mainly positive health for men and women and conclude that their findings support the convergence model that gender equality will be associated with a convergence in health outcomes for men and health [126]. Patriarchal gender norms have been dominant in both Ethiopia and Australia. These have been shown to be both detrimental and protective for different aspects of men’s health. Protective in that men hold most power in a patriarchal society and detrimental in that confirming to a stereotype of heterosexual masculinity may be incompatible with taking care of one’s health or seeking health care. Courtenay [28] (p. 1385) assesses how “men’s social practices that undermine their health are often signifiers of masculinity and instruments that men use in the negotiation of social power and status”. Consequently, when patriarchal norms are challenged through law reform and social change, men may become more willing to engage in health protective behaviours.

Bourdieu maintains that the extent to which people hold capitals determines the power they will have in society. He also sees it as a means by which those who already hold power can reproduce it. Thus, it is an important way in which health inequities, including those between men and women, are maintained. Bourdieu’s capitals assist in ensuring that a class dimension is introduced into the debate on GLED. To a significant extent class sources of power also reflect patriarchal power, which is reflected in the extent to which men control institutions and resources in society. Yet as commentators [20,28] have argued, patriarchy implies a particular dominant and socially acceptable form of heterosexual masculinity. Access to power and capital flows to those who conform to both the dominant class and gender groups. Other groups to a greater or lesser extent are excluded by class or race or sexuality [53]. Thus it is certainly true as Risman et al. [26] argue, that gender can be understood as “a system of inequality embedded in all aspects of society” (p. 19) that has diverse implications for women’s and men’s health experiences. This picture is also strongly influenced by class and access to economic capital. Intersectionality theory (coined by Crenshaw [127]) has increasingly been applied in gender and health research to understand that in each social cultural context gender intersects with other differences and systems of oppression [21,128]. Intersectionality highlights how people’s differences and the causes of inequities risk being masked when reduced to one category. Our conclusions support Robinson’s [129] (p. 70). view that intersectionality is a theory of power relationships, which demonstrates how “various systems of oppression, including racism, sexism, capitalism, and heteropatriarchy intersect and reinforce each other in order to stratify and dominate minority groups” The challenge we have identified in our study is being able to find sufficient empirical data with which to examine each of these intersectionalities, let alone the interactions between them. Figure 3 provides a framework that we hope will be used by researchers in other settings to add richer analysis to our understanding of GLED.

Broader literature has linked power and control to longevity [82] and provides evidence that groups in society with most access to power live the longest. Yet our analysis indicates that overall women hold less economic and political power in both Australia and Ethiopia yet in both countries they live longer. This conundrum leaves many questions unanswered. The CSDH Women and Gender Equity Knowledge Network [113] noted that the resources, power, authority and control that men are more likely to have access to can also be harmful to men’s health despite the many tangible benefits it gives them. Their report [113] (p. xii) noted that these benefits “to men do not come without a cost to their own emotional and psychological health, often translated into risky and unhealthy behaviours, and reduced longevity”. It is also true that socioeconomic gradients operate for both genders confirming the need for gender analysis to be informed by a class analysis. Gender equity is unlikely to play out in the same way in different countries. An analysis [130] of LED in relation to the GDI Index indicates that there was a positive association in Europe and the Americas but in Africa the relationship was negative. These findings suggest that in Europe and the Americas greater gender equality leads to a narrowing of the gender life expectancy gap while in Africa it had the opposite effect. Both these findings and ours suggest more research is required to understand the impact of increased gender equity and life expectancy and that the studies need to consider the context of each country both in terms of its epidemiology and its social, political and economic history and current situation.

Our analysis of Australia and Ethiopia relied on limited data and revealed many data gaps, but has allowed us to provide a partial examination of the ways in which capitals mediate the relationship between social determinants of health and life expectancy and indicate some of the contextual reasons for the GLED. The contrasts between the two countries support the case for more detailed studies that examine the reasons for the differences. We agree with Hill and Friel’s [131] conclusion that, in the future, the commercial determinants of health will continue to exert a powerful influence on the health of women and men and that they deserve much more detailed research. Economic capital is shaped by the commercial determinants of health. This impact happens, for example, through transnational corporations affecting food supplies as a result of trade agreements and aggressive advertising [132]. The impacts are also likely to differ by gender, but research on commercial determinants is it a nascent stage and as it develops it should incorporate a gender lens.

### 4.1. Data Considerations

Analysis of gendered life expectancy is often conducted on aggregate numbers across income groups or for whole countries. Yet there is likely to be much diversity in gendered experiences hidden by aggregate numbers. A Women’s Peace and Security Index [133] measuring women’s rights and opportunities in the United States reveals vast differences between the states, with Massachusetts scoring almost four times better than Louisiana. Hence analyses need to be sensitive to the ways in which aggregate data may mask difference in gender experiences. The data available in both countries reveal gaps in gender data and suggest that more detailed data would be helpful to assist understanding of the complexity of reasons behind gender life expectancy differences. Perez [134] in her book Invisible Women describes the “gender data gap”, which she argues masks many of the experiences of women, especially as data, which can consider the intersectionality between gender and other variables, including class and ethnicity, is even rarer. The importance of good data on gender has been highlighted by the Global Health 50/50 [135]. Such data will permit more sophisticated analysis, which when analysed in light of social theory, will reveal the ways in which social and economic factors combine to determine life expectancy of different groups. Data collections should also include sex and gender data, which includes going beyond binary representation in order to produce results that are inclusive of the full gender spectrum [130]. To fully understand why life expectancy differs between gender, data need to be collected as part of official national statistical data and to ensure that data examines the range of economic, social and cultural; factors that affect health. Currently, the quest to understand the gender differences in life expectancy is very limited by the paucity of good data.

### 4.2. Limitations

This paper has relied on a rapid assessment of health, social and economic position of women in two countries. A more in-depth analysis involving more case countries would have produced a more fine-tuned analysis. This paper has focused on difference in life expectancy. We recognise that the difference behind the gender differences in mortality may be different to those driving life expectancy difference.

Currently most global health data collected reflects a binary notion of gender and excludes statistics on people who are intersex, gender diverse, transgender or non-binary [27]. This illustrates how a dominant gender order marginalizes certain group. In this article we took life expectancy and the gender gap between men and women as a springboard for our examination of gender, power and health, and so acknowledge that the current data masks the experiences of many people.

## 5. Conclusions

While our examination of GLED has enabled us to demonstrate the value of the social determinants’ lens and Bourdieu’s capitals in unpacking some of the factors accounting for the difference between men and women, it has also highlighted the equally important differences between average life expectancies in different countries and between different groups within countries. Comprehensive approaches redress the unequal impact of social, political and commercial determinants of health on countries and marginalized groups within them. Shining a gender lens is vital and contributes to more complex understanding of health inequities and how to reduce them.

## Figures and Tables

**Figure 1 ijerph-18-00661-f001:**
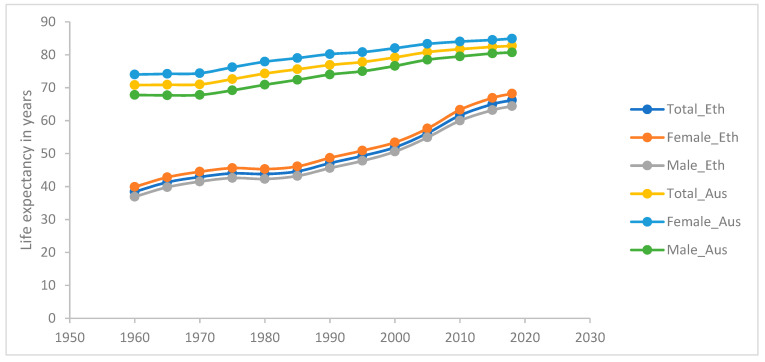
Life expectancy trend by gender in Australia and Ethiopia, 1960–2018.

**Figure 2 ijerph-18-00661-f002:**
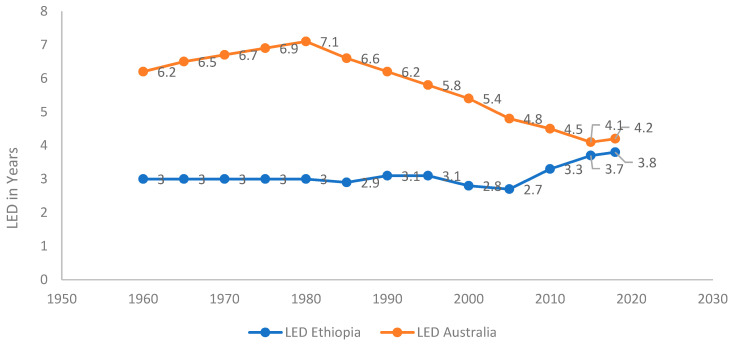
Gendered life expectancy difference (GLED) trend between male and female in Australia and Ethiopia, 1960–2018.

**Figure 3 ijerph-18-00661-f003:**
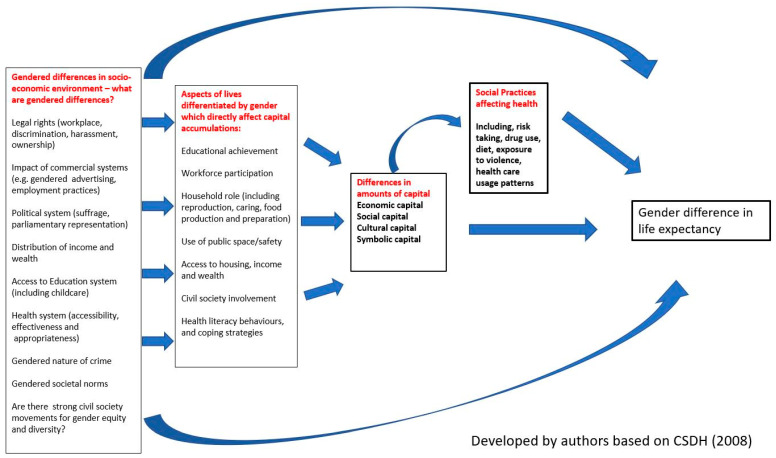
A framework to guide research explaining inequalities in gender differences in life expectancies. CSDH: Commission on the Social Determinants of Health.

**Table 1 ijerph-18-00661-t001:** Life expectancy and life expectancy difference (in years) by country income.

Countries Grouped by Income Level	Life Expectancy, Total, at Birth	Life Expectancy, Female, at Birth	Life Expectancy, Male, at Birth	Gender Life Expectancy Difference
Low-income	63.49	65.41	61.59	3.82
Middle-income	71.86	74.11	69.75	4.36
High-income	80.66	83.35	78.13	5.22
World	72.56	74.87	70.39	4.48

Source: World Bank database, 2018.

**Table 2 ijerph-18-00661-t002:** Top 10 causes of death in Australia and Ethiopia: males and females.

Rank	Cause of Death (Male), Australia, (%)	Cause of Death (Female), Australia, (%)	Cause of Death (Male), Ethiopia, (%)	Cause of Death (Female), Ethiopia, (%)
1st	Coronary heart disease (12.5)	Dementia including Alzheimer disease (11.8)	Infectious and Parasite disease (29)	Infectious and Parasite disease (41)
2nd	Lung cancer (6.1)	Coronary heart disease (9.5)	External cause of death (16) ^1^	Indeterminate [51]
3rd	Dementia including Alzheimer disease (6.1)	Cerebrovascular disease (7.6)	Cardiac related disease (14) ^2^	Cardiac related disease (14)
4th	Cerebrovascular disease (5.1)	Lung cancer (4.7)	Indeterminate (13)	External cause of death (8)
5th	Chronic obstructive pulmonary disease (COPD) (4.6)	Chronic obstructive pulmonary disease (COPD) (4.4)	Neoplasm (6) ^3^	Neoplasm (6)
6th	Prostate cancer (4)	Breast cancer (3.9)	Mental and nervous system disorder (6) ^4^	Renal disorders (4)
7th	Colorectal cancer (3.5)	Colorectal cancer (3.2)	Gastrointestinal disorder (6) ^5^	Mental and nervous system disorder (4)
8th	Diabetes (3.1)	Diabetes (2.8)	Nutritional and endocrine disorder (4) ^6^	Nutritional and endocrine disorder (4)
9th	Suicide (2.8)	Heart failure and complications and ill-defined heart disease (2.4)	Renal disorders (3) ^7^	Gastrointestinal disorder (3)
10th	Cancer of unknown (2)	Influenza and pneumonia (2.2)	Respiratory disorder (1) ^8^	Unspecified (3)

Source: causes of death in Northern Ethiopia (Melaku et al.) [50] and Australian Institute of Health and Welfare (AIHW), National Mortality Database; ^1^ accident, self-harm, assault, war, lack of food; ^2^ hypertension, congestive heart failure, ischemia; ^3^ oesophagus, intestine, cervix, stomach; ^4^ epilepsy, Alzheimer’s, mental disorder; ^5^ chronic liver disease, acute abdomen, gastric ulcer; ^6^ diabetes, anaemia; ^7^ renal failure, kidney disorder; ^8^ Asthma, COPD.

**Table 3 ijerph-18-00661-t003:** Gender equality indexes: GII and GDI.

GII ^1^	2000	2005	2010	2015	2018
Australia	0.160	0.139	0.138	0.110	0.103
Ethiopia		0.614	0.580	0.519	0.508
**GDI ^2^**	**2000**	**2005**	**2010**	**2015**	**2018**
Australia	0.961	0.969	0.975	0.975	0.975
Ethiopia	0.741	0.759	0.817	0.841	0.844

Source: United Nations Development Programme. Gender Inequality Index; United Nations Development Programme. Gender Development Index; ^1^ UN Gender Inequality Index (GII). The GII is an inequality index. It shows the loss in potential human development due to disparity between female and male achievements in three dimensions: (a) reproductive health (maternal mortality ratio; adolescent birth rate); (b) empowerment (female and male population with at least secondary education; female and male shares of parliamentary seats); and (c) labour market participation (female and male participation rates). Overall, the GII reflects how women are disadvantaged in these dimensions. The GII ranges between 0 and 1. Higher GII values indicate higher inequalities between women and men and, thus, higher loss to human development. Available at: http://hdr.undp.org/en/indicators/68606; ^2^ UN Gender Development Index (GDI): the Gender Development Index (GDI) measures gender gaps in human development achievements by accounting for disparities between women and men in three basic dimensions of human development—health, measured by female and male life expectancy at birth; education, measured by female and male expected years of schooling for children and female and male mean years of schooling for adults ages 25 years and older; and command over economic resources, measured by female and male estimated earned income. The GDI is the ratio of the HDIs calculated separately for females. It is a direct measure of gender gap showing the female HDI as a percentage of the male HDI. The GDI is based on the absolute deviation of GDI from gender parity, 1.00. Higher value indicates higher gender parity. Available at: http://hdr.undp.org/en/content/gender-development-index-gdi.

## Data Availability

Not applicable.

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
