# Peer review of "New Perspective on Why Women Live Longer Than Men: An Exploration of Power, Gender, Social Determinants, and Capitals"

_ijerph, 2021, doi:10.3390/ijerph18020661_

Round 1
Reviewer 1 Report
The article deals with a very interesting topic which is the gendered determinants of differences in life expectancy and their explanations, and I think it contributes to the scientific debate, especially Figure 3 which is a very useful theoretical tool.
However, in my view, there are two main flaws that need to be fixed before publication:
- since gender is one of the key elements to explain GLED, I would have expected a wider discussion on the sociological (but not only) literature coming from gender studies and, in particular, some reflections about the social construction of masculinity and health. I think this insight could make both the section 3.2 and the discussion more empirically grounded and theoretically relevant.
This point is similar to a point made by Reviewer 3 regarding the potential value of further elaboration on patriarchy. In response we have included extended the background section to include a more substantive discussion of literature on patriarchy and the social construction of masculinity and health. Reference to the following literature has been included:
Drever, F., Doran, T., & Whitehead, M. (2004). Exploring the relation between class, gender, and self rated general health using the new socioeconomic classification. A study using data from the 2001 census. Journal of Epidemiology and Community Health, 58, 590–596
Hankivsky, O. (2012). Women’s health, men’s health, and gender and health: implications of intersectionality. Social Science & Medicine, 74, 1712–1720.
Springer KW, Hankivsky O, Bates LM. Gender and health: relational, intersectional, and biosocial approaches. Soc Sci Med. 2012 Jun;74(11):1661-6. doi: 10.1016/j.socscimed.2012.03.001. Epub 2012 Mar 20. PMID: 22497844.
Connell, R. (2012). Gender, health and theory: conceptualizing the issue, in local and world perspective. Social Science & Medicine, 74, 1675–1683
Courtenay, W. H. (2000). Constructions of masculinity and their influence on men’s well-being: A theory of gender and health. Social Science & Medicine, 50(10), 1385–1401
Jocalyn Clark, Richard Horton, A coming of age for gender in global health, The Lancet, Volume 393, Issue 10189, 2019, Pages 2367-2369, https://doi.org/10.1016/S0140-6736(19)30986-9.
Sarah Hawkes, Pascale Allotey, As Sy Elhadj, Jocalyn Clark, Richard Horton, The Lancet Commission on Gender and Global Health, The Lancet, Volume 396, Issue 10250, 2020, Pages 521-522, https://doi.org/10.1016/S0140-6736(20)31547-6.
S Hawkes, K Buse. The politics of gender and global health
C McInnes, K Lee, J Youde (Eds.), The Oxford handbook of global health politics, Oxford University Press, New York (2020), pp. 237-264
We also return to this theme in the discussion as noted by tracked changes
- It is unclear to me the choice of the two empirical cases. I think it is not sufficient to say that they are "two countries well-known to one or other of the authors" (p. 3, 101-102). Even if the section 3.2 is well articulated and shows similarities and differences between the two Countries, I think some methodological justifications should be provided.
|
We have added a stronger justification: “Given the novelty of the examination of GLED and the anticipated challenges of our reliance on diverse secondary data sources we decided we would only be able to manage a robust albeit preliminary analysis of two cases. Our selection criteria for the two were a combination of theoretical and methodological: Theoretically we decided we would gain greater analytical advantaged by comparing two countries with: (i) Diverging national income levels: one high and one low income; and (ii) higher than expected overall LE given level of GDP. Methodologically, we selected two countries for which relevant literature and/or data on policies, and political and historical context was readily available to us and members of the author team had knowledge of these sources |
- I suggest to merge the section of the discussion with the conclusion.
We have left the conclusion as a final summary seems helpful and the other two reviewers did not request this change
Author Response
A point-by-point response is attached below.

Reviewer 2 Report
This article shows a comparative analysis of secondary data, with the aim of knowing in depth how life expectancy behaves in people according to gender (female and male). The study carries out an analysis that takes as a reference macro social determinants of health and access to the four capitals of Bourdieu (economic, cultural, social and symbolic) in two countries, Australia and Ethiopia. It is analyzed how these dynamics can translate into differences in life expectancy by gender. So, how access to these four capitals differentiate the experiences of men and women in the two countries and ways in which they either expand or contract individuals’ life chances. The study through the Bourdieu Capitals framework demonstrates the many factors that must be taken into account to determine health and lifestyle impacts.
See response to Reviewer 1 above
The article is written and designed appropriately and presents its results in a meaningful way for the hypothesis raised. The study design is replicable with these and other countries, as explained in the research. The research will attract readers due to the significance of the topic on life expectancy and the incidence of gender in its trajectory. However, I think it would be very positive to include in more detail why to analyze these two countries and not others, or how the analysis based on these two countries can help the research in this regard.
I consider that the research question is well defined and makes a relevant contribution. The research has a correct design, very descriptive and correlation analysis. The study provides more information and documentation regarding this issue and the relevance it may have on the analysis of the development of health as a right for all people. The reference to economic, cultural, social and symbolic capitals has been useful to highlight a series of factors that can influence life expectancy and also to consider the interactions between different forms of capital. However, I believe that it is a descriptive analysis and that it would be positive if it were to introduce a social impact approach now or in the future. I consider it important to introduce in future, analyzes a methodological approach such as the communicative methodology. It introduces the identification of elements of transformation and elements of exclusion, to identify ways in which individual and collective agency can transform structural and even biological determinants and influence an increase in health and life expectancy. Figure 3 provides a representation of how these influences work to affect life expectancy. I think it can be a useful tool to examine the gendered life expectancy difference, but I think that the Social Practices affecting health could have a crucial relevance of the GLED results. I believe that the authors could value placed them in line with Bourdieu's capitals between “Differences in amount of capital” and “Gender difference in life expectancy”. Perhaps in this way, the analysis can answer one of the questions that the study leaves open: in general, women have less economic and political power in both Australia and Ethiopia, but in both countries they live longer. This is only a reflection that the authors can value if they consider it convenient to do so.
We thank the reviewer for these thought-provoking comments. We have introduced more theoretical material on both the “capitals” and “gender” theories and attempted to strengthen the link between the theoretical frameworks and the secondary data sources we have analysed. We have now reviewed the following references in the background:
Laberge, S. (1995). Toward an Integration of Gender into Bourdieu’s Concept of Cultural Capital, Sociology of Sport Journal, 12(2), 132-146. Retrieved Dec 16, 2020, from https://journals.humankinetics.com/view/journals/ssj/12/2/article-p132.xml
Abel, T. (2008). Cultural capital and social inequality in health. Journal of Epidemiology & Community Health, 62(7), e13
Filippo Oncini & Raffaele Guetto (2018) Cultural capital and gender differences in health behaviours: a study on eating, smoking and drinking patterns, Health Sociology Review, 27:1, 15-30, DOI: 10.1080/14461242.2017.1321493
Gareth Wiltshire, Jessica Lee & Oli Williams (2019) Understanding the reproduction of health inequalities: physical activity, social class and Bourdieu’s habitus, Sport, Education and Society, 24:3, 226-240, DOI: 10.1080/13573322.2017.1367657
Anna Schneider‑Kamp. Health capital: toward a conceptual framework for understanding the construction of individual health
Shim (2010) Cultural Health Capital: A Theoretical Approach to Understanding Health Care Interactions and the Dynamics of Unequal Treatment
We hope that the reviewer feel that has made the paper less descriptive. We agree that in the future a more sophisticated explanatory analysis will be important, but we note that this will require the collection of new empirical data. We argue that the paper is presenting exploratory rather than descriptive research in that it seeks to determine if what is being observed might be explained by a novel combination of existing social science theories. In doing so we believe the paper is providing a basis for future empirically driven explanatory research, but this will require the collation of new data.
Author Response

(The authors gave the same response as above.)

Reviewer 3 Report
This manuscript takes a close look at the gendered life expectancy difference (GLED), aiming to employ a Bourdieusian framework to be able to integrate social, cultural, and economic determinants.
My main concern is that the manuscript does not sufficiently engage with the existing and growing body of literature on Bourdieusian forms of capital in a health context. The authors are advised to engage at least with the following articles. The first - Shim (2010) - is relevant as she focuses on how cultural (and to a certain degree social) capital influence health inequality. The second one - Schneider-Kamp (2020) - could provide a theoretical foundation through its conceptualization of “health capital” as an aggregation of health-related forms of capitals, allowing to sharpen the ideas of this manuscript regarding how forms of capitals can have explanatory value regarding GLED. The second article would also allow to deepen the Bourdieusian perspective regarding symbolic forms of capital, which are not used correctly in the current manuscript.
Shim JK. Cultural health capital: A theoretical approach to understanding health care interactions and the dynamics of unequal treatment. Journal of Health and Social Behavior. 2010;51:1–15.
Schneider-Kamp A. Toward Health capital: toward a conceptual framework for understanding the construction of individual health. Social Theory and Health. 2020. http://doi.org/10.1057/s41285-020-00145-x
|
We thank the reviewer for the suggested references. We have now noted the he concept of “Cultural health capital” and linked it to our data in terms of its likely impact on health service use. We have extended the discussion of the existing applications of Bourdieu’s capitals in the background. This discussion includes the reference to the following: Laberge, S. (1995). Toward an Integration of Gender into Bourdieu’s Concept of Cultural Capital, Sociology of Sport Journal, 12(2), 132-146. Retrieved Dec 16, 2020, from https://journals.humankinetics.com/view/journals/ssj/12/2/article-p132.xml Abel, T. (2008). Cultural capital and social inequality in health. Journal of Epidemiology & Community Health, 62(7), e13Filippo Oncini & Raffaele Guetto (2018) Cultural capital and gender differences in health behaviours: a study on eating, smoking and drinking patterns, Health Sociology Review, 27:1, 15-30, DOI: 10.1080/14461242.2017.1321493Gareth Wiltshire, Jessica Lee & Oli Williams (2019) Understanding the reproduction of health inequalities: physical activity, social class and Bourdieu’s habitus, Sport, Education and Society, 24:3, 226-240, DOI: 10.1080/13573322.2017.1367657Anna Schneider‑Kamp. Health capital: toward a conceptual framework for understanding the construction of individual healthShim (2010) Cultural Health Capital: A Theoretical Approach to Understanding Health Care Interactions and the Dynamics of Unequal Treatment |
Shim (2010) Cultural Health Capital: A Theoretical Approach to Understanding Health Care Interactions and the Dynamics of Unequal Treatment
If the authors are willing to engage more closely with the above literature, it could help to develop the manuscript towards a contribution that could be described as the “gendering of health capital”.
Detailed comments and suggestions:
Line 110: “determine the ways in which patriarchy operates” - where did you establish that there is a patriarchy at? this needs further elaboration.
See response to first point of Reviewer 1
Line 121: having “access to these four capitals” is an awkward formulation. agents in the social field possess capitals rather than just having access to them. here, integrating your theoretical framework the suggested literature (see above) would improve the conceptual rigor of your manuscript.
We have reworded this paragraph to reflect the reviewer’s point that capitals are possessed. We have also responded to this comment by including more literature on social capital in the background section.
Line 316: please elaborate on which interpretation of Bourdieu’s theories you are using here to bring things to the micro level. also, what do you mean by “fabric of everyday life”? are you referring to “everyday practices”?
We have removed the reference to micro and replaced “fabric of everyday life”? with “everyday practices
Line 382: please elaborate - why would higher economic capital translate to longer life expectancy? money can be used towards better healthcare, but it can also be used to buy cigarettes, alcohol, fast cars and other detrimental goods.
We have added four references to support the link between economic capital and life expectancy (new ref 88-91)
Line 385: this statement is questionable. cultural capital is much more than “just” literacy, education and qualifications. please engage with the literature suggested above.
We have revised the text to encompass a broad view of cultural capital and drawn on Shin’s article to do this (see p. 17 Lines 1-3 and 10-12)
Lines 405ff: your interpretation of social capital is closer to Bourdieu’s concept than economic and cultural capital. nevertheless, this section would profit from engaging with some additional literature regarding the effect of social capital on health and life expectancy, at least including the following two:
Turner B. Social capital, inequality and health: The Durkheimian revival. Social Theory & Health. 2003;1:4–20.
Kennelly B, O’Shea E, Garvey E. Social capital, life expectancy and mortality: A cross-national examination. Social Science & Medicine. 2003;56:2367–2377.
We have added a reference to the work of Turner at the start of section on social capital (p. 20 Li 1-6). We haven’t quoted Kennelly et al as they found no relationship between social capital and health. We suggest that this is because of the very narrow definition of social capital they used (levels of trust in others and membership in voluntary organisations). There isn’t room in the paper to provide a discussion of differing definitions of social capital
Line 443: NO - you cannot say that symbolic capital “reflects the accumulation of the other three forms of capital”.
We have removed this sentence
Line 450: replace “hasn’t” by “has not” - similar with other contractions throughout the manuscript.
Line 467: saying that capitals “influence” everyday practices is very imprecise. please use a more precise and conceptually rigorous lanuage to make your point.
We have reformulated this sentence to read as follows:
The four types of capitals, singly and in combination strongly shape the social health practices that have a direct impact on mortality
Lines 468-471: this makes sense from an intuitive point of view (except as noted in my next point). but to turn this into an academically sound argument, you need to either expand on your understanding of the forms of capital and their influence on health and health-related practices OR (preferably) built on existing conceptualizations of health capital (see the literature mentioned above).
|
Thank you for the suggestion that we might use Schdeider-Kamp’s concept of health capital. We have added the following to the paragraph highlighted here: “The ways in which Bourdieu’s capitals overlap and affect health has led Schneider-Kamp to propose the notion of ‘health capital’ as a concept able to aid understanding of contemporary trends in health practices and health discourse. This concept seems most helpful in relation to the extent to which people can adopt healthy social practices to influence their health and in order to identify the structural factors that impede adopting them. A stark example in our two countries is that in Australia nearly all communities can take the provision of safe drinking water for granted. In Ethiopia its provision for many still depends on individual agency to seek and collect the water – often a task that falls to women”. |
Lines 470: how does social capital influence acceptability?
Social capital can influence acceptability (the only mention in the article is in section 3.3.) as more extensive social capital may mean that people are exposed to a broader range of ideas.
Lines 491-492: this is where the manuscript might contribute to the literature, in particular Schneider-Kamp (2020). demonstrating and arguing for that health capital is strongly gendered.
Thank you for the suggestion that we might use Schdeider-Kamp’s concept of health capital. We have added the following to the paragraph highlighted (above)
Line 540: remove the figure OR explain all the arrows and elaborate on the “Differences ...” and “Social ...” boxes.
|
We have kept the figure as we agree with reviewer 1 who commented of our paper “ I think it contributes to the scientific debate, especially Figure 3 which is a very useful theoretical tool”. We have, however, as suggested provided more explanation of Figure 3. As follows: “It shows that gendered differences in socio-economic environments shape the ways in which men and women are able to develop capabilities and accumulate the different capitals. This results in strong gender effects on the amounts of the different capitals individuals accumulate and these directly affect GLED. The extent of capitals accumulated also affects the social practices individuals adopt. These practices are strongly shaped by gender norms and also contribute to GLED” We have also made minor amendments to the figure so a new version is included in the paper. We have also added mention of Figure 3 to the abstract |
Author Response

(The authors gave the same response as above.)

Round 2
Reviewer 1 Report
The Authors did a significant effort to take into consideration the reviewers' comments and I think the quality of the article has improved.
The references section is not alphabetically ordered, I suggest to fix it.
Reviewer 3 Report
The authors made an effort of engaging more with the literature on capitals and health. This has strengthened the scientific soundness of the manuscript and provides grounds for a reassessment of the contributions of the paper.
There are some issues with the new references that needs to be sorted out, though:
1) References 38 and 41 are identical - they should be merged.
2) Reference 38/41 needs to be referred to on Page 22, Line 15 (after "health has led Schneider-Kamp").
3) The order of the references is weird. It is not by where they are mentioned in the document nor alphabetical nor chronological. Please choose the right order for the reference style you are using.
